# Post-COVID-19 syndrome: Physical capacity, fatigue and quality of life

Sebastian Beyer[1]*, Sven Haufe[1], Meike Dirks[2], Michèle Scharbau[3], Viktoria Lampe[1], Alexandra Dopfer-Jablonka[4], Uwe Tegtbur[1], Isabell Pink[3], Nora Drick[3], Arno Kerling[1]

1 Department of Rehabilitation and Sports Medicine, Hannover Medical School, Hannover, Germany, 2 Clinic for Neurology, Hannover Medical School, Hannover, Germany, 3 Department of Respiratory Medicine, Hannover Medical School, Hannover, Germany, 4 Clinic for Rheumatology and Immunology, Hannover Medical School, Hannover, Germany

☯ These authors contributed equally to this work.

* beyer.sebastian@mh-hannover.de

**Data Availability Statement:** You can access the study data through the following links: https://mhh-publikationsserver.gbv.de/receive/mhh_mods_00002583 or https://doi.org/10.26068/mhhrpm/20231010-000.

## Abstract

### Purpose

Post-Covid-19 syndrome is defined as the persistence of symptoms beyond 3 months after severe acute respiratory syndrome coronavirus 2 (SARS-CoV-2) infection. The most common symptoms include reduced exercise tolerance and capacity, fatigue, neurocognitive problems, muscle pain and dyspnea. The aim of our work was to investigate exercise capacity and markers of subjective wellbeing and their independent relation to post-COVID-19 syndrome.

### Patients and methods

We examined a total of 69 patients with post-COVID-19 syndrome (23 male/46 female; age 46±12 years; BMI 28.9±6.6 kg/m$^2$) with fatigue and a score $\geq$22 in the Fatigue Assessment Scale (FAS). We assessed exercise capacity on a cycle ergometer, a 6-minute walk test, the extent of fatigue (FAS), markers of health-related quality of life (SF-36 questionnaire) and mental health (HADS).

### Results

On average the Fatigue Assessment Scale was 35.0±7.4 points. Compared with normative values the VO2max/kg was reduced by 8.6±5.8 ml/min/kg (27.7%), the 6MWT by 71±96 m (11.9%), the health-related quality of life physical component score by 15.0±9.0 points (29.9%) and the mental component score by 10.6±12.8 points (20.6%). Subdivided into mild fatigue (FAS score 22–34) and severe fatigue (FAS score $\geq$35), patients with severe fatigue showed a significant reduction of the 6-minute walk test by 64±165 m (p<0.01) and the health-related quality of life physical component score by 5.8±17.2 points (p = 0.01). In multiple regression analysis age (β = −0.24, p = 0.02), sex (β = 0.22, p = 0.03), mental (β = −0.51, p<0.01) and physical (β = −0.44, p<0.01) health-related quality of life and by trend the 6-minute walk test (β = −0.22, p = 0.07) were associated with the FAS.

**Funding:** YES - The study was supported and funded by Erwin Röver Foundation. The funders had no role in study design, data collection and analysis, decision to publish, or preparation of the manuscript.

**Competing interests:** NO authors have competing interest.

## Conclusion

Patients with post-COVID-19 syndrome show reduced maximal and submaximal physical performance as well as limitations in quality of life, particularly pronounced in the physical components. These results are essentially influenced by the severity of fatigue and implicating the need for targeted treatments.

## Introduction

The post-acute sequelae of severe acute respiratory syndrome coronavirus 2 (SARS-CoV-2) infection—also referred to as post-COVID-19 –is estimated to be prevalent in at least 65 million individuals worldwide [1]. According to the WHO post-COVID-19 syndrome (PCS) is defined as the persistence of symptoms beyond 3 months after severe acute respiratory syndrome coronavirus 2 (SARS-CoV-2) infection [2]. Approximately 80% of hospitalized and non-hospitalized patients have symptoms lasting longer than 12 months [3]. Important risk factors associated with PCS appears to be pre-existing illnesses (especially cardiorespiratory, autoimmune and oncological previous diseases, and neuropsychiatric disorders), female gender, respiratory symptoms at onset, length of hospital stay and severity of illness, although milder courses also can trigger PCS [1, 4–6]. The most common symptoms that contribute mainly to decreased health recovery and reduced resilience include fatigue, reduced exercise tolerance and muscular strength, neurocognitive problems, muscle pain and dyspnea [7, 8].

Physical activity and exercise capacity are significant predictors of long-term survival and both are reduced in the context of PCS [9–12]. In contrast good physical performance to a certain extent protects against a severe course of SARS-CoV-2 infection and reduces the likelihood of hospitalization [8, 13]. In addition, there is a strong relation between physical performance and health-related quality of life (HrQoL), and exercise training is used as a therapeutic tool for a variety of physical and mental disorders [14–16]. Mental and physical components of HrQoL are reduced after acute SARS-CoV-2 infection as well as in PCS patients [17], both by the disease itself and by public health measures such as quarantine [18, 19].

However, the relative negative impact of PCS on physical function, fatigue, mental conditions and their associations are not fully elucidated. This knowledge is vital for a targeted treatment because until now a specific causal therapy for PCS patients does not exist. We aimed to identify potential predictors that contribute to PCS-related fatigue and thereby might help to better understand the relative significance of physical and mental risk factors on their association with PCS severity. We conducted a thorough assessment of patients after SARS-CoV-2 infection and diagnosed PCS and analyzed associations of fatigue severity with parameters of physical capabilities, quality of life and mental conditions in order to be able to implement an optimal therapy strategy in the long term.

## Methods

### Study design and participants

The current evaluation was a cross-sectional analysis of data collected between September 2021 and June 2022. The study was conducted at a university medical hospital in the city of Hannover (Lower Saxony, Germany) with individuals diagnosed with PCS. We included patients from October 2021 to May 2022 who consulted the pneumological post-COVID-19 outpatient clinic of Hannover Medical School, which is the first point of contact for almost all patients with suspected post-COVID19 syndrome, regardless of the various possible symptoms

that could lead to the diagnosis of PCS. Patients were referred to our university hospital by general practitioners or pneumologists due to persisting symptoms ≥3 months after SARS--CoV-2 infection.

According to pre-study defined criteria, we included female and male volunteers aged 18 years or older who reported a 3-month continuing impairment of capability after COVID-19 (detection by PCR) infection with a Fatigue Assessment Scale (FAS) score ≥22 points. Exclusion criteria were current participation in another intervention study, clinically relevant acute or chronic infections, pregnancy, preceding surgery less than 8 weeks ago, joint replacement that is less than 6 months old, tumor -associated diseases in the last 5 years such as any illnesses or functional impairments which preclude participation in a physical training intervention.

The Clinic for Rehabilitation and Sports Medicine at Hannover Medical School in cooperation with the Department of Respiratory Medicine was responsible for the study design, statistical planning, inclusion of study participants, data collection and analysis, as well as the preparation of the manuscript. This study was carried out in accordance with the Declaration of Helsinki and is registered on German Clinical Trials Register (registration number: DRKS0002793). The institutional ethics review board of Hannover Medical School approved the study (No.9822_BO_S_2021) and written informed consent was obtained prior to the inclusion of participants.

## Assessments

After study inclusion, all subjects completed a comprehensive medical evaluation including pulmonary function testing measured by body plethysmography standardized to European Respiratory Society (ERS). We assessed height and weight in a standardized fashion and estimated fat and fat-free mass with a bioimpedance analysis (InBody 720, Biospace, Seoul, Korea). To determine steps per day, patients received a wearable (Forerunner 45, Garmin, Olathe, United States).

## Maximal and submaximal exercise testing

For testing parameters of exercise capacity and maximum power output patients performed an incremental exercise test using a spirometric system (Oxycon CPX, CareFusion, Würzburg, Germany) on a speed independent bicycle ergometer (Ergoline P150, Bitz, Germany) with 60 to 70 revolutions per minute. The exercise test were carried out in the presence of a physician (specialist for cardiology or internal medicine) together with a sports scientist or a medical technical assistant, all experienced and trained in conducting, evaluating and interpreting exercise performance diagnostics. Except six examinations the incremental test started with a load of 20 Watt (W) increasing in 10 W steps every minute and was stopped with the onset of subjective overexertion because of peripheral muscle fatigue and/or pulmonary limitations (four started with a load of 50 W increasing in 10 W steps every minute, two started with a load of 50 W increasing in 16.67 W steps every minute). The subjective perceived exertion was assessed by the Borg-scale [20]. Heart rate and oxygen uptake were continuously measured breath by breath. Body weight normalized values for maximum power output and $VO_2max$ were also expressed as percentage to age- and gender-adjusted reference values [21, 22].

The 6MWT is a test to assess submaximal exercise capacity by walking distance. Participants walked a slope-free corridor for a total of six minutes at their own speed and the reached distance was recorded in meters [23]. The tests were supervised by a trained sports scientists or medical technical assistants.

## Questionnaires

We distributed a questionnaire for the estimation of HrQoL (short form 36 [SF-36]). The SF-36 using eight subscales, each with a scale ranging from 0 to 100, culminating in two summated scales, the mental and physical component score. A higher mental and physical component score means a better HrQol. In addition, we calculated the mean of the eight subscales for classification of our patients in comparison with a systematic review [24].

We assessed the severity of depression and anxiety with the Hospital Anxiety and Depression Scale (HADS) [25]. Scores for the anxiety and depression subscales range from 0 to 21, higher scores indicating more severe anxiety or depression [25]. Fatigue was measured with the FAS (© FAS Fatigue Assessment Scale: ild care foundation [http://www.ildcare.nl]). We divided the patients into two groups, the subdivision between fatigue (n = 32) and extreme fatigue (n = 37) is a FAS score ≥35 points.

## Statistical analysis

Data were first tested for normal distribution and variance homogeneity with Kolmogorov-Smirnov test and the Levene test, respectively. Differences between baseline parameters were compared by Student t test for unpaired samples, the Mann-Whitney u test or the chi-square test, respectively. Univariate associations between parameters were tested using Pearson's correlation coefficient or Spearman's correlation coefficient. Differences between subgroups of fatigue and extreme fatigue (based on FAS score) were analyzed by Student t test for unpaired samples or the Mann-Whitney u test. Differences between subgroups regarding vaccination status were tested with a one-way ANOVA and Bonferroni´s *post hoc* test.

Finally, a stepwise backward multivariate linear regression was performed to identify parameters associated with PCS severity. The type-I-error was set to 5% (two-sided). All statistical analyses were performed with IBM SPSS 27 Statistics (IBM Corporation, New York, USA). Unless otherwise stated, values are given as mean ± SD or as median [min; max].

## Results

Between September 2021 and June 2022, 77 patients were recruited for study examinations. Of those, eight have to be excluded as they did not meet the in-/ exclusion criteria. Four withdrew their consent to participate in the study, one did not appear for examinations, one was too young, one was unable to confirm a positive PCR/antigen test, and one had a FAS score of less than 22 points. Finally, 69 patients with PCS were included in the analysis.

## Participants' characteristics

Anthropometric and clinical characteristic of patients are shown in Table 1. Women and men did differ for body weight, body fat, $FEV_1$, $FEV_{ex}$ and the FAS score (Table 1). Pre-existing illnesses were cardiovascular disease (n = 1), depression (n = 6), diabetes (n = 5), hypertension (n = 15), COPD (n = 6), restrictive lung disease (n = 2), renal insufficiency (n = 1), and also 20 use tobacco. 21 of the patients have no pre-existing illnesses. Patients were taking antihypertensives (n = 14), ß-blockers (n = 6), antiobstructive medications (n = 16), antidepressants (n = 7), antidiabetics (n = 3), anticoagulants (n = 3), thyroid hormones (n = 11) and 25 patients were not taking any medications. Out of 69 patients two were not vaccinated, 30 once, 26 twice, and 11 thrice vaccinated against SARS-CoV-2 at study inclusion. Regarding vaccination status there were no differences between vaccination subgroups in physical performance, fatigue severity, and mental or physical component scores of HrQol.

**Table 1. Subject characteristics at baseline.**

| | Total group | women | men | p-value |
|---|---|---|---|---|
| Subjects (women/men) | 69 | 46 | 23 | 0.006 |
| Age (years) | 46 ± 12 | 46 ± 11 | 46 ± 12 | 0.820 |
| Body weight (kg) | 85.2 ± 22.4 | 79.2 ± 23.4 | 97.3 ± 14.4 | <0.001 |
| Body mass index (kg/m$^2$) | 28.5 [11.1] | 25.7 [12.1] | 29.2 [9.0] | 0.114 |
| Body fat (%) | 32.1 ± 11.6 | 35.2 ± 11.6 | 25.6 ± 8.6 | <0.001 |
| weeks since Covid-19 diagnosis | 43 [32] | 43 [30] | 36 [42] | 0.378 |
| steps per day | 7159 [3641] | 6934 [3570] | 7444 [3397] | 0.445 |
| maximum power output (watt/kg) | 1.67 [0,90] | 1.67 [0.95] | 1.70 [0.80] | 0.653 |
| VO2max (ml/min/kg) | 22.5 ± 6.4 | 21.9 ± 6.4 | 23.8 ± 6.3 | 0.248 |
| 6MWT (m) | 525 ± 88 | 516 ± 83 | 543 ± 96 | 0.265 |
| max HR ergometry (bpm) | 150 ± 22 | 153 ± 21 | 145 ± 23 | 0.188 |
| Breathing reserve (%) | 39 [20] | 37 [23] | 43 [17] | 0.408 |
| FEV1 (ml) | 3055 [970] | 2910 [580] | 3655 [985] | <0.001 |
| FVCex (ml) | 3983 ± 1043 | 3495 ± 776 | 4892 ± 857 | <0.001 |
| max Lactat concentration (mmol/l) | 5.12 ± 2.19 | 5.12 ± 2.42 | 5.12 ± 1.67 | 0.995 |
| Borg-scale | 19 [1] | 19 [1] | 19 [1] | 0.405 |
| FAS score | 35.0 ± 7.4 | 36.6 ± 6.7 | 32.0 ± 8.0 | 0.024 |
| QoL physical score | 35.2 ± 9.0 | 34.6 ± 8.9 | 36.4 ± 9.4 | 0.478 |
| Qol mental score | 40.9 ± 12.8 | 40.6 ± 13.9 | 41.6 ± 10.8 | 0.738 |
| HADS depression score | 6.0 [7.0] | 5.0 [7.0] | 8.0 [6.0] | 0.279 |
| HADS anxiety score | 6.5 [8.0] | 6.0 [9.0] | 7.0 [7.0] | 0.714 |

Differences between groups were analyzed with *Students t-test* for unpaired samples or the Mann-Whitney test, data are mean ± SD or median [interquartile range].

## Exercise capacity and health-related quality of life

The exercise capacity when expressed to age- and sex-specific norm values, was 72.3±18.5% predicted for VO2$_{max}$, 88.1±16.1% predicted for the 6-min walking distance, 99.6±18.7% predicted for FEV$_1$ and 104.1±18.5% predicted for FVC$_{ex}$ (see also Fig 1 for individual data).

The mental component score and the physical component score for HrQoL were below norm values (79.4±24.9% predicted; 70.1±18.0% predicted, respectively) (for more detailed data see Figs 2 and 3 and Table 2 for parameters divided into fatigue and extreme fatigue).

## Association of PCS severity and health-related outcomes

The FAS score was correlated to the 6MWT (r: -0.459, p<0.001), to the HrQoL physical component score (r: -0.366, p = 0.003), to the HrQoL mental score (r: -0.410, p<0.001), to the HADS depression score (r: 0.331, p = 0.009) and the HADS anxiety score (r: 0.290, p = 0.022). The FAS score was correlated only by trend to maximum power output (r:-0.225, p = 0.067) and to VO2$_{max}$ (r: -0.228, p = 0.064).

In a stepwise backward multivariate linear regression analysis with the FAS score as dependent variable and age, sex, time since diagnosis of PCS, VO2$_{max}$, 6MWT, HrQoL physical and mental component score, HADS depression and anxiety score as independent variables, sex (β = 0.220, p = 0.025), age (β = -0.236, p = 0.022), the HrQoL physical component score (β = −0.440, p<0.001), the HrQoL mental component score (β = −0.512, p<0.001) and by trend the 6MWT (β = −0.215, p = 0.068) predicted the FAS score. The model explained 53.2% of the total variation in PCS severity (FAS score).

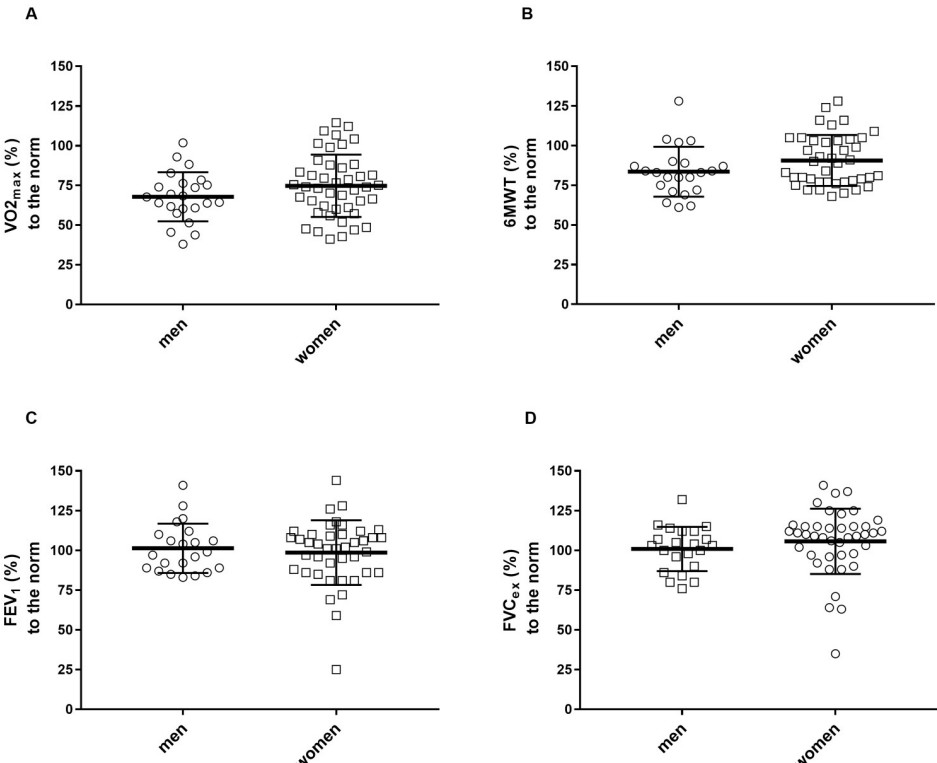

**Fig 1. Exercise capacity (A: VO2$_{max}$; B: 6MWT) and pulmonary capacity (C: FEV$_1$; D: FVC$_{ex}$) in relation to the norm values.** Data are individual values with mean ± SD.

## Discussion

We investigated anthropometric and clinical features of PCS patients and their association to PCS fatigue severity. Main finding of the study is that patients with PCS show reduced maximal and submaximal physical performance as well as limitations in quality of life, particularly pronounced in the physical components (especially in the physical role function). These results are essentially influenced by the severity of fatigue.

In addition to the risk factors mentioned above, obesity is also one of the predictors for the development of PCS [26], and BMI as well as body fat percentage were also elevated in our patient population. Obesity often is associated with a low cardiorespiratory fitness which also increases the risk for a severe course of COVID-19 as well as for the development of PCS [13].

As in other studies, our patient population showed a reduced VO2$_{max}$ as well as a reduced walking distance in the 6MWT [27–29]. The performance in percentage was more limited in the maximal ranges than in the submaximal range, although the FAS correlated significantly only with the 6MWT performance, a finding also observed in other studies [6], and tended to correlate with VO2$_{max}$. Possible causes of impaired cardiopulmonary fitness can be, in addition to reduced fitness even before the onset of the disease, a further reduction due to inactivity and a sedentary lifestyle because of COVID-19 related symptoms [29] or government-ordered measures of quarantine [30, 31]. Further reasons might include COVID-19-induced physical impairments, side effects of drug treatment or a combination of these factors [32].

The leading cause for discontinuing the exercise test in our patients was peripheral exhaustion (n = 45). Seeßle at al. [3] describe that about 40% of PCS patients still suffer from dyspnea

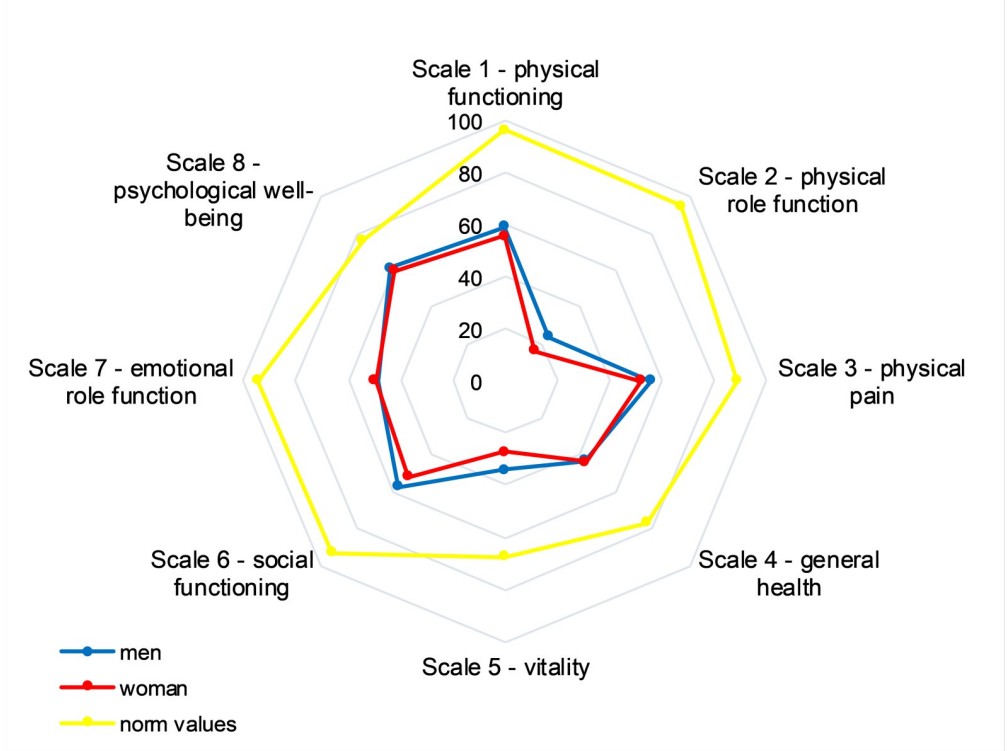

**Fig 2. Subscales of the HrQoL questionnaire for men and women compared to the norm values.**

after about one year. In our study, this was not observed in this high number either at rest (Fig 1) or during spiroergometric exercise testing. During spiroergometry, we detected specific pulmonary limitations in two patients with chronic obstructive pulmonary disease and in two patients with restrictive breathing patterns. Although SARS-CoV-2 vaccination has been observed to reduce the risk of PCS, with two vaccinations appearing to be more effective than one [33] and the majority of our examined patients had been vaccinated at least once, the severity of PCS-related outcomes was not influenced by vaccination status in our cases.

Interestingly, although most of the patients subjectively reached a high level of self-perceived exertion at the BORG scale, in some cases there was no corresponding increase in heart rate, breathing rate and lactate concentration, which argues against cardiopulmonary or metabolic exhaustion in this context. Since this occurred predominantly in patients with a high FAS score we assume that central factors are partly limiting exercise capacity in these patients. In fact, mental fatigue could lead, due to increased cerebral adenosine activity amongst other factors, to a stronger perception of exertion during exercise [34]. This assumption is supported by Gibson et al. [35] who found no limitations in regard to muscle function in patients with chronic fatigue syndrome and demonstrated normal muscle physiology before exercise and during recovery. In addition these patients had increased perceived exertion scores during exercise in relation to heart rate and blood lactate concentration [35, 36], a finding similar to that in our PCS patients with high FAS scores. However, more recent research suggests that impaired muscle function could also contribute to decreased exercise capacity in patients with PCS [11, 37], which demonstrates the need for further research on this area.

Independent of COVID-19, post-infectious fatigue is a common phenomenon that can last even longer than 6 months after various infections (e.g. Epstein-Barr virus). Data on the

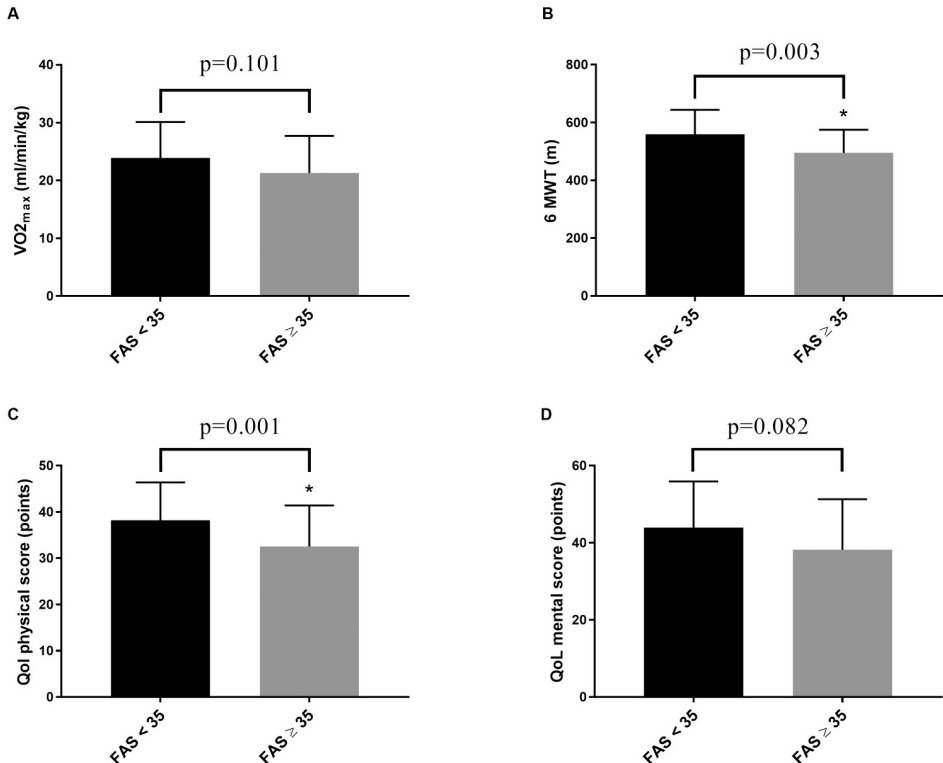

**Fig 3. Exercise capacity (A: VO2$_{max}$; B: 6MWT) and HrQoL (C: Physical component score; D: Mental component score) divided into mild and severe fatigue based on FAS score.** The framed p-values are given for differences between mild and severe fatigue as analyzed by Student t test for unpaired samples. Data are mean ± SD.

incidence of fatigue after COVID-19 infection show that the number of affected patients varies widely, ranging up to 34% after hospitalization [38, 39]. Our study samples reflects the composition of the general population with and without pre-existing illness. Most of those illnesses in our study cohort are known for limiting quality of life and physical performance to certain extents (e.g. depressive symptoms, diabetes, hypertension). We cannot consider with certainty the degree of influence of the pre-existing illness on our observations regarding reduced physical performance and health-related quality of life relative to norm values. However, because

**Table 2. Subscales of the HrQoL questionnaire for men and women compared to the norm values.**

|  | men | women | men (% of the norm) | women (% of the norm) |
|---|---|---|---|---|
| Scale 1—physical functioning | 59.1±21.0 | 55.5±21.0 | 61.4 | 57.7 |
| Scale 2—physical role function | 23.9±31.3 | 16.5±22.8 | 25.0 | 17.3 |
| Scale 3—physical pain | 56.1±25.0 | 52.3±25.6 | 63.1 | 58.9 |
| Scale 4—general health | 43.1±14.8 | 43.9±17.3 | 55.9 | 56.9 |
| Scale 5—vitality | 34.1±16.7 | 27.2±15.5 | 50.5 | 40.3 |
| Scale 6—social functioning | 57.4±32.0 | 51.8±27.0 | 61.6 | 55.7 |
| Scale 7—emotional role function | 48.5±45.7 | 49.6±44.8 | 51.3 | 52.5 |
| Scale 8—psychological well-being | 61.6±18.4 | 59.6±23.1 | 80.6 | 77.9 |

Data are mean ± SD.

the reference values for these outcomes were also assessed from the general population, a major influence on study outcomes is unlikely.

Quality of life is known to be reduced in patients with PCS [29, 40]. However, there is controversy over which items are most affected. A systematic review in which 7 studies using the SF-36 questionnaire in PCS patients found lowered mean values for the total scores ranging from 52.7–79.96, whereas our result with 45.7 was clearly below the lower range. Indeed, an inclusion criteria for this study was a high FAS score, suggesting a pre-selection of our patients. While in five studies of the review [24] the subscale physical function (scale 1) had the best value, in three studies of the review in a recent cross-sectional study [41] and also in our study, the physical role (scale 2) was most affected. In contrast to us, Maes et al. [42] could not observe any impairment in the area of social HrQoL in PCS patients. A multicenter study that had investigated the influence of obesity on various PCS symptoms on average found somewhat lower values for depression and anxiety severity than we did [43], although our mean values were still within a normal range. Our observed associations between fatigue severity, physical performance and quality of life may be attributed to the interlinked nature of these parameters. In general, physical exercises leads to an improvement in physical well-being and mood. The lack of physical activity can be secondary to PCS-related fatigue, which in turn lead to a reduced formation of serotonin, endorphin, brain-derived neurotrophic factor, and vascular endothelial growth factor, leading to restricted neurogenesis, which in turn results in reduced neuronal plasticity with the consequence of increased depression, anxiety and reduced self-esteem [44]. From these interactions a vicious cycle can manifest, which makes a therapy of PCS even more complicated.

In the case of an assessment for returning to work not only the physical performance but also significantly reduced quality of life and fatigue should always be considered. In addition, health-related quality of life is closely linked with fatigue and other physio-somatic symptoms [39], therefore a multimodal therapy concept should be offered. Treatment therefore should be symptom-oriented and consider the required medical specialties e.g. pneumology, cardiology, neurology or psychosomatic medicine. Our data, that demonstrate the interlinkage of physical, mental and social aspects of health, supports the note that exercise should be one part within a multimodal treatment concept considering reduced mobility, breathlessness, stress, and problems with mood, memory, attention and tasks of daily living. The training program could be based on the WHO recommendations [45], taking into account a patient's individual resilience, mobility, and any other health condition. The WHO recommendation also points out that exercise should incorporate endurance, strength, but also balance and breathing techniques. Finally, overloading due to physical exercises training should be prevented. Thus, in a state of extreme fatigue, avoiding strenuous exercise is crucial to prevent post-exertional malaise or a crash [1]. According to the study of Davis et al. triggers for relapses can be inadequate exercise, physical or mental activity in relation to the extent of the disease as well as stress and after 7 months suffering from PCS more than half of the patients were found to be unable to work or only to a limited extent [46].

Our study has strengths and limitations. The strengths are that we investigated physical performance (by means of spiroergometry and on basis of the lactate performance curve) as well as quality of life and fatigue and their relationship to each other. The major limitation of the study is the relatively small sample number. This allows only results with high effect sizes to be recorded with sufficient accuracy and also limit the generalizability of the findings, particularly in such a heterogeneous sample of patients. Pre-COVID-19 state of health was not known and the extent of the deterioration thus not clearly definable. Furthermore, the study was performed in a German, mainly Caucasian population and, therefore, may not have sufficient generalizability to other nonwestern populations or ethnicities. Accordingly the reported

findings of our study should be interpreted with caution and further large-scale studies are needed to verify the observed results.

## Conclusions

Patients with fatigue due to PCS show impaired physical performance and quality of life. These three items seem to be interdependent and influence each other. Our findings support the need for further investigations in regard to multimodal therapy concepts which should be adapted to the individual's performance in order to reduce fatigue and therefore to improve physical capacity and quality of life.

## Acknowledgments

The results of this study are presented clearly, honestly, and without fabrication, falsification, or inappropriate data manipulation. We thank the patients for their study participation.

## Author Contributions

**Conceptualization:** Meike Dirks, Uwe Tegtbur, Nora Drick, Arno Kerling.

**Data curation:** Sebastian Beyer, Nora Drick.

**Formal analysis:** Sebastian Beyer, Sven Haufe, Isabell Pink.

**Funding acquisition:** Uwe Tegtbur, Arno Kerling.

**Investigation:** Sebastian Beyer, Meike Dirks, Michèle Scharbau, Viktoria Lampe, Isabell Pink, Nora Drick, Arno Kerling.

**Methodology:** Sven Haufe, Alexandra Dopfer-Jablonka, Arno Kerling.

**Project administration:** Sebastian Beyer.

**Resources:** Alexandra Dopfer-Jablonka.

**Supervision:** Sven Haufe, Uwe Tegtbur, Arno Kerling.

**Visualization:** Sebastian Beyer.

**Writing – original draft:** Sebastian Beyer, Sven Haufe, Arno Kerling.

**Writing – review & editing:** Sebastian Beyer, Sven Haufe, Meike Dirks, Michèle Scharbau, Viktoria Lampe, Alexandra Dopfer-Jablonka, Uwe Tegtbur, Isabell Pink, Nora Drick, Arno Kerling.

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
