## [Decision Letter · Decision Letter 0]

6 Sep 2023

PONE-D-23-20678Post Covid19 syndrome: physical capacity, fatigue and quality of lifePLOS ONE

Dear Dr. Beyer,

Thank you for submitting your manuscript to PLOS ONE. After careful consideration, we feel that it has merit but does not fully meet PLOS ONE’s publication criteria as it currently stands. Therefore, we invite you to submit a revised version of the manuscript that addresses the points raised during the review process.

We look forward to receiving your revised manuscript.

Kind regards,

Kalyana Chakravarthy Bairapareddy, PhD

Academic Editor

PLOS ONE

Journal Requirements:

   "YES - The study was supported and funded by Erwin Röver Foundation."

Additional Editor Comments:

The authors are requested to provide detailed justifications for the reviewers comments, improve the flow of information and resubmit the manuscript with the suggested modifications.

Reviewers' comments:

Reviewer's Responses to Questions

**Comments to the Author**

1. Is the manuscript technically sound, and do the data support the conclusions?

Reviewer #1: Partly

Reviewer #2: Partly

2. Has the statistical analysis been performed appropriately and rigorously? 

Reviewer #1: No

Reviewer #2: Yes

3. Have the authors made all data underlying the findings in their manuscript fully available?

Reviewer #1: Yes

Reviewer #2: Yes

4. Is the manuscript presented in an intelligible fashion and written in standard English?

Reviewer #1: Yes

Reviewer #2: No

5. Review Comments to the Author

Reviewer #1: As a reviewer, I have analyzed the manuscript titled "Post Covid19 syndrome: physical capacity, fatigue and quality of life." Below is a critical analysis of each section:

1. Title:

The title of the manuscript is concise and clearly reflects the focus of the study. It includes the relevant keywords, making it easy for readers to understand the content. However, one minor suggestion would be to capitalize "COVID-19" consistently throughout the title for better readability.

2. Abstract:

The abstract provides a brief overview of the purpose, methods, and main findings of the study. It is well-structured and communicates essential information. However, it could be improved by including specific numerical results rather than stating that "VO2max/kg was reduced by 27.7%, the 6MWT by 11.9%, the health-related quality of life physical component score by 29.9%," etc. Quantitative values would make the abstract more informative.

3. Introduction:

The introduction is well-written and provides a comprehensive background on Post-COVID syndrome, its symptoms, and potential risk factors. It cites relevant literature and highlights the importance of understanding the impact of PCS on physical function, fatigue, and mental conditions. However, it would benefit from more recent citations, as the latest research on PCS might be available after the manuscript's knowledge cutoff date in September 2021.

4. Methods:

The methods section is sufficiently detailed to understand the study design and procedures. It provides inclusion and exclusion criteria, the number of participants, and the assessment tools used. However, there are a few areas that need clarification. The criteria for selecting participants should be further justified, and any potential biases in recruiting patients from a pneumological post-COVID-19 outpatient clinic should be discussed. Additionally, it would be helpful to include information on the training and expertise of those conducting the tests to ensure the validity of the results.

5. Results:

1. Sample Size: The study included 69 patients, which might be considered relatively small for some analyses. A larger sample size would strengthen the generalizability of the findings and allow for more robust statistical analyses.

2. Exclusion Criteria: It is unclear why eight patients were excluded from the analysis based on the in-/exclusion criteria. The reasons for their exclusion should be explicitly stated to ensure transparency.

3. Pre-existing Illnesses: The number of patients with specific pre-existing illnesses is relatively small, making it challenging to draw conclusive inferences regarding the impact of each condition on PCS. Additional discussion of these conditions' potential influence on the study's outcomes would provide a more comprehensive understanding.

4. Statistical Significance: While correlations between FAS scores and various health-related outcomes are provided, these associations' significance levels (p-values) are missing. Reporting the statistical significance is crucial for interpreting the strength and validity of these relationships.

6. Discussion:

1. Interpretation of Results: The discussion provides a detailed interpretation of the study's main findings, linking fatigue severity to reduced physical performance and quality of life. However, the potential causal relationships and underlying mechanisms behind these associations should be further explored and discussed.

2. Comparison to Previous Studies: The discussion compares the study's results to previous research on PCS and fatigue. However, it would be helpful to include more recent studies, as the latest literature might provide additional insights that could be relevant to the current analysis.

3. Limitations: The study acknowledges certain limitations, such as the relatively small sample size, lack of information on participants' pre-COVID-19 health, and potential generalizability issues to other populations. While these limitations are acknowledged, it would be beneficial to further discuss their potential impact on the study's findings and conclusions.

4. Clinical Implications: The discussion briefly touches on the need for multimodal therapy concepts to address PCS-related fatigue and its impact on physical capacity and quality of life. However, a more extensive discussion of potential treatment strategies and their practical application for patients would enhance the manuscript's clinical relevance.

The results section presents the study's findings effectively, but some missing statistical information should be addressed. The discussion provides a valuable interpretation of the results, but it could be strengthened by incorporating more recent literature and delving deeper into potential mechanisms and clinical implications. Additionally, a more comprehensive exploration of the study's limitations and their impact on the conclusions would enhance the manuscript's overall rigor and validity.

Reviewer #2: Introduction:

The introduction section is lacking a strong rationale with respect to the study. Authors can work a bit more on literature synthesis.

Methodology:

Methods section was not properly adhered and reported as per the standard guidelines. The study design was unclear.

Why authors did not consider about vaccination status of the participants recruited ? as it might effect the physical health status and performance.

Results & Discussion: The reported findings and explanation found satisfactory.

6. PLOS authors have the option to publish the peer review history of their article (what does this mean?). If published, this will include your full peer review and any attached files.

Reviewer #1: **Yes: **Ravi Shankar REDDY

Reviewer #2: No

---

## [Author Response · Author response to Decision Letter 0]

29 Sep 2023

Reviewer 1

Reviewer #1:As a reviewer, I have analyzed the manuscript titled "Post Covid19 syndrome: physical capacity, fatigue and quality of life." Below is a critical analysis of each section:

1. Title:

The title of the manuscript is concise and clearly reflects the focus of the study. It includes the relevant keywords, making it easy for readers to understand the content. However, one minor suggestion would be to capitalize "COVID-19" consistently throughout the title for better readability.

Response: Done (title and short title on title page)

2. Abstract:

The abstract provides a brief overview of the purpose, methods, and main findings of the study. It is well-structured and communicates essential information. However, it could be improved by including specific numerical results rather than stating that "VO2max/kg was reduced by 27.7%, the 6MWT by 11.9%, the health-related quality of life physical component score by 29.9%," etc. Quantitative values would make the abstract more informative.

Response: We added the numerical data to the abstract (see lines 15-17).

3. Introduction:

The introduction is well-written and provides a comprehensive background on Post-COVID syndrome, its symptoms, and potential risk factors. It cites relevant literature and highlights the importance of understanding the impact of PCS on physical function, fatigue, and mental conditions. However, it would benefit from more recent citations, as the latest research on PCS might be available after the manuscript's knowledge cutoff date in September 2021.

Response: We agree with the reviewer and conducted a new search for the latest literature on PCS. We included new citations and incorporated these literatures in our introduction (see tracked-changes [blue-coloured and underlined] for new references and text).

4. Methods:

The methods section is sufficiently detailed to understand the study design and procedures. It provides inclusion and exclusion criteria, the number of participants, and the assessment tools used. However, there are a few areas that need clarification. The criteria for selecting participants should be further justified, and any potential biases in recruiting patients from a pneumological post COVID19 outpatient clinic should be discussed. Additionally, it would be helpful to include information on the training and expertise of those conducting the tests to ensure the validity of the results.

Response: Since almost all post-COVID-19 patients at our university hospital were primarily examined in the pneumology outpatient clinic, recruitment for our study also took place there. Because the pneumonological clinic is the first point of contact, regardless of the various possible symptoms of post-COVID-19, we do not assume a bias in recruiting patients for our study in PCS patients. We added a short description in the methods to point this out (see lines 64 – 68). The expertise of the investigators was added in lines 96 - 99 and 109 - 110.

5. Results:

1. Sample Size: The study included 69 patients, which might be considered relatively small for some analyses. A larger sample size would strengthen the generalizability of the findings and allow for more robust statistical analyses.

Response: The reviewer is right. A larger sample size would have been desirable in this rather heterogenic sample of individuals to reduce the variability, and enable a higher explanatory power of the statistical analysis. Therefore, we already named the relatively small sample size as major limitation in our study (see line 282 - 283). As also suggested in a later comment of the reviewer we added a short text passage to make this limitation more clear (line 284 - 285 and 288 - 289).

2. Exclusion Criteria: It is unclear why eight patients were excluded from the analysis based on the in-/exclusion criteria. The reasons for their exclusion should be explicitly stated to ensure transparency.

Response: We agree with the reviewer and added the reasons for their exclusion to ensure transparency (see lines 138-141).

3. Pre-existing Illnesses: The number of patients with specific pre-existing illnesses is relatively small, making it challenging to draw conclusive inferences regarding the impact of each condition on PCS. Additional discussion of these conditions' potential influence on the study's outcomes would provide a more comprehensive understanding.

Response: In fact, most of the pre-existing illnesses in our study participants are known to reduce both quality of life and physical performance. However, those illnesses are typical in the general population reflecting the normal composition of our study sample in terms of the inclusion of apparently healthy and non-healthy individuals. We agree with the reviewer that this issue should be addressed in the discussion and added a brief text section on the potential influence of the pre-existing illnesses on our outcomes (see lines 228 – 231).

4. Statistical Significance: While correlations between FAS scores and various health-related outcomes are provided, these associations' significance levels (p-values) are missing. Reporting the statistical significance is crucial for interpreting the strength and validity of these relationships.

Response: Perhaps the reviewer missed it, but we have already stated significance levels for the correlations between FAS scores and various health-related outcomes (see lines 178-182).

6. Discussion:

1. Interpretation of Results: The discussion provides a detailed interpretation of the study's main findings, linking fatigue severity to reduced physical performance and quality of life. However, the potential causal relationships and underlying mechanisms behind these associations should be further explored and discussed.

Response: We added some text passages to discuss possible mechanism for our observed relationships between parameters of fatigue; well-being and physical performance (see lines 205 – 208, 228 – 231, 254 – 261)

2. Comparison to Previous Studies: The discussion compares the study's results to previous research on PCS and fatigue. However, it would be helpful to include more recent studies, as the latest literature might provide additional insights that could be relevant to the current analysis.

Response: We searched for the latest literature on PCS. We included recent studies and incorporated and discussed them in our discussion section (see tracked new references [blue-colored and underlined].

3. Limitations: The study acknowledges certain limitations, such as the relatively small sample size, lack of information on participants' pre-COVID-19 health, and potential generalizability issues to other populations. While these limitations are acknowledged, it would be beneficial to further discuss their potential impact on the study's findings and conclusions.

Response: We agree, particularly for such a heterogeneous cohort of patients the results should be interpreted with caution and in an optimally confirmed (or rejected) in larger samples. We added a text section to make this clear (see lines 288 – 289).

4. Clinical Implications: The discussion briefly touches on the need for multimodal therapy concepts to address PCS-related fatigue and its impact on physical capacity and quality of life. However, a more extensive discussion of potential treatment strategies and their practical application for patients would enhance the manuscript's clinical relevance.

Response: We added a brief discussion regarding this issue (see lines 265 – 275).

The results section presents the study's findings effectively, but some missing statistical information should be addressed. The discussion provides a valuable interpretation of the results, but it could be strengthened by incorporating more recent literature and delving deeper into potential mechanisms and clinical implications. Additionally, a more comprehensive exploration of the study's limitations and their impact on the conclusions would enhance the manuscript's overall rigor and validity.

Response: As we understand, the reviewer summarizes his remarks made before in this comment. As responded earlier we added some statistical informations (lines 128 - 131), searched and incorporated latest research on PCS and discussed possible mechanisms and treatments options for PCS patients (lines 214 - 217, 222 - 224, 228 – 231, 254 – 261, 265 - 275). We also addressed the impact of the study limitations on the stated conclusions and validity of or study (lines 284 – 285, 288 – 289).

We again thank the reviewer for the constructive and helpful comments on our work and hope the reviewer find improvement with the made revisions.

Reviewer 2

Introduction:

The introduction section is lacking a strong rationale with respect to the study. Authors can work a bit more on literature synthesis.

Response: Thank you for this comment. We extended the introduction on order to point out more clearly the rationale and aim of our current study (see lines 50-55)

We also searched for recent literature on PCS and incorporated these in the introduction to provide additional information on latest research on this field (see blue marked and underlined references).

Methodology:

Methods section was not properly adhered and reported as per the standard guidelines. The study design was unclear.

Response: According to the reviewers comment we overworked our study description based on the appropriate guideline for cross-sectional studies (STROBE). In particular, we specified/ extended our descriptions regarding the study design, setting, locations and dates of inclusion and data collection (lines 62 - 68). In this context, we also added further information on exercise testing (lines 96 – 99, 109 - 110) and statistical analyses (128 – 131).

Why authors did not consider about vaccination status of the participants recruited? as it might effect the physical health status and performance.

Response: The reviewer is right. To state the vaccination status in these patients is important and was forgotten to include in the original manuscript. Out of 69 PCS patients 2 were not vaccinated, 30 once, 26 twice, and 11 thrice vaccinated. To test possible differences between subgroups of vaccinated and non-vaccinated participants we compared physical performance and fatigue severity between subgroups. Using a one-way ANOVA with Bonferroni´s post hoc test, we did not observe differences for physical performance (VO2max, 6-MWT) and the physical health status (fatigue (FAS), health-related quality of life (SF-36)) when comparing groups regarding vaccination status. 

Data here for the reviewer only:

VO2max: not vaccinated (1770 ± 70 ml/min), vaccinated once: (1838 ± 384 ml/min), vaccinated twice: (1709 ± 544 ml/min), vaccinated thrice: (2036 ± 776 ml/min); p=0.390.

6-MWT: not vaccinated (529 ± 28 m), vaccinated once: (528 ± 106 m), vaccinated twice (539 ± 85 m), vaccinated thrice: (475 ± 104 m); p=0.258.

FAS score: not vaccinated (33.0 ± 1.0 points); vaccinated once: (36.5 ± 7.7 points), vaccinated twice (33.6 ± 7.2 points), vaccinated thrice: (34.0 ± 7.1 points); p=0.389.

SF-36 physical component score: not vaccinated (32.2 ± 2.1 points), vaccinated once: (34.7 ± 10.3 points), vaccinated twice: (37.3 ± 9.8 points), vaccinated thrice: (35.6 ± 7.1 points); p=0.577

SF-36 mental component score: not vaccinated (42.6 ± 5.8 points), vaccinated once: (38.3 ± 12.6 points), vaccinated twice: (41.1 ± 12.3 points), vaccinated thrice: (41.9 ± 15.9 points); p=0.419.

In the manuscript, we now included the vaccination status and the finding of no differences regarding the main outcomes between vaccination subgroups in the results section at line 153 – 156.

We also added the following statement to point this out in the discussion section (line 214 - 217).

“Although SARS-CoV-2 vaccination has been observed to reduce the risk of PCS, with two vaccinations appearing to be more effective than one (Byambasuren O et al., BMJ Med., 2023, DOI: 10.1136/bmjmed-2022-000385) and the majority of our examined patients had been vaccinated at least once, the severity of PCS-related outcomes was not influenced by vaccination status in our cases.”

Results & Discussion: 

The reported findings and explanation found satisfactory.

Response: Thank you. In the context of comments of the other reviewer, we extended the discussion on latest literature and added brief discussed on further aspects of our data.

---

## [Editor Report · Decision Letter 1]

3 Oct 2023

Post-COVID-19 syndrome: physical capacity, fatigue and quality of life

PONE-D-23-20678R1

Dear Dr. Sebastian Beyer,

We’re pleased to inform you that your manuscript has been judged scientifically suitable for publication and will be formally accepted for publication once it meets all outstanding technical requirements.

Kind regards,

Kalyana Chakravarthy Bairapareddy, PhD

Academic Editor

PLOS ONE

---

## [Editor Report · Acceptance letter]

12 Oct 2023

PONE-D-23-20678R1 

Post-COVID-19 syndrome: physical capacity, fatigue and quality of life 

Dear Dr. Beyer:

I'm pleased to inform you that your manuscript has been deemed suitable for publication in PLOS ONE. Congratulations! Your manuscript is now with our production department. 

Kind regards, 

on behalf of

Dr. Kalyana Chakravarthy Bairapareddy 

Academic Editor

PLOS ONE